# Recent Advances in Water-Soluble Vitamins Delivery Systems Prepared by Mechanical Processes (Electrospinning and Spray-Drying Techniques) for Food and Nutraceuticals Applications—A Review

**DOI:** 10.3390/foods11091271

**Published:** 2022-04-27

**Authors:** Sílvia Castro Coelho, Berta Nogueiro Estevinho, Fernando Rocha

**Affiliations:** LEPABE—Laboratory for Process Engineering, Environment, Biotechnology and Energy, Department of Chemical Engineering, Faculty of Engineering, University of Porto, Rua Dr Roberto Frias, 4200-465 Porto, Portugal; silviac@fe.up.pt (S.C.C.); frocha@fe.up.pt (F.R.)

**Keywords:** microencapsulation, water-soluble vitamins, electrospinning, electrospraying, spray drying

## Abstract

Water-soluble vitamins are essential micronutrients in diets and crucial to biochemical functions in human body physiology. These vitamins are essential for healthy diets and have a preventive role against diseases. However, their limitations associated with high sensitivity against external conditions (temperature, light, pH, moisture, oxygen) can lead to degradation during processing and storage. In this context, microencapsulation may overcome these conditions, protecting a biomolecule’s bioavailability, stability, and effectiveness of delivery. This technique has been used to produce delivery systems based on polymeric agents that surround the active compounds. The present review focuses on the most relevant topics of water-soluble vitamin encapsulation using promising methods to produce delivery vehicles—electrohydrodynamic (electrospinning and electrospraying) and spray-drying techniques. An overview of the suitable structures produced by these processes is provided. The review introduces the general principles of the methods, advantages, disadvantages, and involved parameters. A brief list of the used physicochemical techniques for the systems’ characterization is discussed in this review. Electrospinning and spray-drying techniques are the focus of this investigation in order to guarantee vitamins’ bioaccessibility and bioavailability. Recent studies and the main encapsulating agents used for these micronutrients in both processes applied to functional food and nutraceutical areas are highlighted in this review.

## 1. Introduction

Nowadays, functional foods are becoming essential for modern consumers. In fact, the incorporation of food bioactive ingredients in our diet has an impact on society’s health and well-being because it provides benefits and minimizes the risk of disease. These ingredients include proteins, minerals, vitamins, phytochemicals, polyphenols, probiotics, carotenoids, and flavors [1]. However, one of the challenges in the advancement of functional foods, food packaging, and preservation is the bioactive compounds preservation in terms of stability and bioactivity. These limitations compromise their benefits. In addition, the unfavorable flavor, low solubility and stability, poor bioavailability, and possible reaction with other product components are undesired effects that restrict the uses of these ingredients and their advantages [2].

In order to overcome the external environmental conditions (light, oxygen, moisture, temperature), mask organoleptic characteristics, and allow the controlled release of the ingredients over time, microencapsulation has been used in industry and is the focus of continuous research for food packaging, functional foods, nutraceutical, and pharmaceutical applications.

Microencapsulation is a production process of small structures produced by polymeric walls that surround bioactive compounds (Figure 1) [3,4].

The produced particles can be presented with a diameter ranging from 1 μm to 1000 μm. The microstructures provide several advantages such as protection against enzymatic degradation and easy administration [5]. It allows for maximizing the controlled release rate and bioavailability of compounds during the processing and storage of formulated products. The formation of a barrier to involve and protect the ingredients from the external environment permits avoiding changes and deterioration in the food systems and gastrointestinal tract [3,6]. It allows for the development of new products with better nutritional quality. Based on the viable manipulation of process parameters, it is possible to fabricate and optimize microstructures with a high specificity of physicochemical characteristics according to the application field [7].

Several microencapsulation techniques have been reported for the entrapment of bioactive compounds such as vitamins into matrices. Microencapsulation techniques implicate steps such as the preparation, homogenization, and atomization of the solution; and dehydration of the atomized formulated systems [8]. The structures are composed of a solid, liquid, or gaseous material and the microencapsulated compound (core) is entrapped into a microencapsulating agent denominated by a “wall”. Microencapsulation techniques are classified into two categories: chemical and physical [9,10,11]. The methods, shown in Table 1, are based on chemical reactions and solvents that present some disadvantages when compared with physical methods such as residual organic solvents present in the structures, and reduction in the activity and stability of the compounds.

Spray-drying, liposomes, spray chilling, freeze-drying, solvent evaporation, melt extrusion, complex coacervation, fluidized bed coating, ionic gelation, and electrohydrodynamic techniques are the methods used for effective microencapsulation of food ingredients [12]. Table 2 summarizes the main advantages and disadvantages of these techniques.

The preservation of biocompounds and their functional activity and characteristics can be improved by using these techniques. Among these methods, electrohydrodynamic and spray-drying procedures are the most promising, attractive, and simple encapsulation methods used for the production of micro/nanostructures and entrapment of bioactive compounds (Table 2).

Biopolymers such as chitosan, gum arabic (GA), alginate, maltodextrin, starch, zein, whey protein, or combined polymers are useful materials for entrapment and have been used by the scientific community as wall materials in the food and nutraceutical industries due to their biodegradability and low toxicity (Figure 2) [24,25].

This review provides an overview of the recent state of the art of the potential encapsulation of water-soluble vitamins using electrohydrodynamic and spray-drying techniques applied to food and nutraceutical areas.

## 2. Water-Soluble Vitamins

Vitamins are essential micronutrients in diets that ensure the biochemical functions of the human body and prevent diseases [26]. They act as antioxidants, hormones, and mediators for cell signaling, cell/tissues regulators, and differentiation [27]. They are sensitive compounds that are degraded during cooking and storage processes by factors such as light, heat, oxygen, moisture, pH, time, and reducing agents [28]. Consequently, vitamin encapsulation can overcome limitations associated with external agents such as oxidants, heat, and low solubility, and promotes effective delivery into the body.

Water-soluble and fat-soluble vitamins are two main groups of this type of micronutrient. Water-soluble vitamins are important for growth, development, and human body function. However, due to physiological conditions, it is not possible to keep them for a long period of time in the body [29]. B-complex group and vitamin C are the main compounds with degradation activity from food processes. Therefore, factors such as bioavailability and bioaccessibility should be considered in current research. Commonly, water-soluble vitamins show a pH sensitivity with bioaccessibility reduction when exposed to small intestine pH values [30].

Microencapsulation techniques such as electrospinning have been the focus of investigation for these types of vitamins. In fact, the non-use of high temperature appeared as a crucial alternative to overcome these limitations associated with these compounds. Another important factor to be considered is the effect of the hydrophobicity/hydrophilicity of the wall materials on the encapsulation method. In fact, the protection of water-soluble vitamins is more efficient for a hydrophobic shell preventing the diffusion of vitamins into food until the desired timepoint [31].

New approaches have been investigated in order to achieve processes with lower temperature and duration [29].

### 2.1. B-Complex Group

The B-complex group comprises micronutrients that act as cofactors for enzymatic reactions in metabolism processes. These compounds are vitamin B1 (thiamine), vitamin B2 (riboflavin), vitamin B3 (niacin), vitamin B5 (pantothenic acid), vitamin B_6_ (pyridoxine, pyridoxal, and pyridoxamine), vitamin B9 (folic acid), and vitamin B12 (cobalamin). They cannot be stored in the human body and should be consumed daily [32]. These vitamins present high sensitivity to light, pH conditions, and temperature. Therefore, encapsulation is an important step in minimizing vitamins’ bioavailability and increasing their effectiveness in situ [33].

#### 2.1.1. Vitamin B1

Vitamin B1 (vitB1), also known as thiamine, is a coenzyme precursor [34]. It is essential for carbohydrate metabolism and plays a vital function in the cardiovascular system as well as in the nervous, immune, and muscular systems [32]. As an essential micronutrient, vitB1 needs to be consumed daily due to the incapacity of the human body to store B12 [35]. VitB1 is found in yeast, grains, and meat and in lower amounts in vegetables and fruits. VitB1 presents reduced stability against environmental conditions such as temperature, pH, humidity, oxygen, and metal ions [35]. Carlan et al., reported the successful encapsulation of vitB1 using different encapsulating agents by a spray-drying method [35]. In terms of achieved stability, the best results were developed for chitosan and modified chitosan presenting low vitB1 loss.

#### 2.1.2. Vitamin B2

Riboflavin, also known as vitamin B2 (vitB2), is a coenzyme in several redox reactions [34]. VitB2 is present in the liver, milk, dark-green leafy vegetables, enriched bread, and cereal. VitB2 is a photosensitive vitamin that converts amino acids into niacin [32]. VitB2 has functions in oxidation–reduction reactions in the body and the cellular system; regeneration and growth of tissues [32,36].

#### 2.1.3. Vitamin B3

Niacin is the generic name of the active form of vitamin B3 (vitB3), also known as nicotinic acid or nicotinamide. It is a water-soluble vitamin present in food products such as meat, poultry, and fish, being accessible as a supplement. Niacin is synthesized in the body and is converted to NAD+ [37]. NAD+ is crucial in different phases, namely its reduction to NADH. NAD and NADH are essential for the production of energy, cholesterol, and fats; and the synthesis/repair of DNA [34]. This vitamin is sensitive and loses its activity with shelf life and storage. VitB3 deficiency in the diet is the cause of pellagra nutritional disease [38].

#### 2.1.4. Vitamin B5

Vitamin B5 (vitB5) is a water-soluble B-complex vitamin, also known as pantothenic acid or pantothenate [27]. VitB5 is a component of an important coenzyme A and an essential vitamin for energy metabolism that allows the conversion of carbohydrates, fats, and proteins into energy [27]. Moreover, vitB5 is necessary for the production of the brain neurotransmitter acetylcholine, responsible for memory and reducing anxiety [39].

#### 2.1.5. Vitamin B6

Vitamin B6 (vitB6) is a water-soluble vitamin necessary in several reactions including amino acid metabolism, glycogen, and different enzymatic reactions concerning immune and growth functions [32]. VitB6 also has an important function in the synthesis of neurotransmitters [40]. It appears in several forms such as pyridoxine, pyridoxal, and pyridoxamine. VitB6 is present in meats, poultry, fish, grain products, fruits, and vegetables [40]. This vitamin is thermally unstable [41]. Modifications to the digestive system, depression, marks of skin aging, and eczema are related to its deficiency [42].

#### 2.1.6. Folic Acid (Vitamin B9)

Folic acid is an essential water-soluble micronutrient that is crucial for several physiological functions in the human body. Also known as vitamin B9 (vitB9), it presents an important function in preventing neural tube defects in children and can decrease the probability of developing vascular diseases, cancer, and Alzheimer’s disease [43]. Folic acid acts on DNA, RNA, and amino acids, supporting rapid cell division [43]. It is not synthesized in the human body but is found in small quantities in fruits, vegetables, and cereals [43]. Nevertheless, the degradation of these vitamins occurs when exposed to light, temperature, oxygen, and acid/alkaline conditions. In fact, this vitamin is one of the most sensitive to losses during food storage and processing due to its chemical reactivity [44]. The microencapsulation of folic acid is a solution to avoid its degradation and maintain its stability and bioactivity in different food matrices. Different matrix materials have been developed for encapsulation of this micronutrient, e.g., spray-drying and electrohydrodynamic techniques, in order to be used as a supplement or food fortification [45].

#### 2.1.7. Vitamin B12

Vitamin B12 (vitB12), or cyanocobalamin, is a cobalt-containing compound (corrinoid) [46]. VitB12 is a stable compound in an aqueous solution of pH 4–7 and does not lose activity when heated at 120 °C [47]. VitB12 is synthesized by bacteria and is found in foods derived from animals such as meats, milk and derivates, and eggs. VitB12 is fundamental in human growth and development. This vitamin plays an important role in the metabolism of fatty acids and aliphatic amino acids; in the brain and nervous system activity; and in blood formation [48]. VitB12 deficiency can lead to anemia, Alzheimer’s disease, neural tube defects, and ulcers [4].

### 2.2. Vitamin C

Vitamin C (vit C), also known as ascorbic acid, is a very common water-soluble vitamin present in fruits and vegetables such as broccoli, citrus fruits, strawberries, tomatoes, raw cabbage, and leafy greens [49]. It cannot be stored in the human body and is not synthetized in it [37]. This micronutrient is a food supplement and a powerful antioxidant [50]. VitC herewith vitamin E, vitamin A, and selenium are four of the main antioxidants recognized by the US Food and Drug Administration [28]. It acts on the synthesis of collagen protein, wound healing, healthy immune and nervous system, quenching or stabilizing the free radicals involved in degenerative diseases, cardiovascular cancer, cataracts, and the immune system [51].

Vit C has the capacity to donate hydrogen atoms to neutralize free radicals and reactive oxygen species that can damage the DNA [26,51]. This vitamin can also prevent the adverse effects of chemotherapeutic agents [52]. However, it is a thermolabile compound [53]. The environmental factors and dissolution in water contribute to minimizing its stability; for example, vitamin degradation at room temperature [26]. This micronutrient is extremely sensitive and can interact with other food ingredients. It is highly reactive and oxidases in the presence of metals such as copper and iron [51]. The microencapsulation technique allows increasing its stability, preserving it from thermo-oxidative degradation. The encapsulation of this vitamin is essential for industry, in order to improve its applicability in food processes. Uddin et al., reported the successful use of several polymers to coat vitC by the spray-drying technique, masking the color and taste changes and controlling its release [54]. Trindade et al., suggested the stability of vitC encapsulated by spray drying, using rice starch and GA encapsulation agents [55].

## 3. Spray Drying

Spray drying is the most used technique for the fabrication of microencapsulation structures due to its simplicity, quickness, and scale-up potential (Table 2). It is an important technique used in the food industry to protect bioactive compounds against light, temperature, and oxidation allowing for good stability of the produced structures. This method involves several steps as shown in Figure 3.

The solution or emulsion is prepared and the ingredients are added and homogenized. The mixture is fed into the spray-dryer equipment and atomized with a nozzle or spinning wheel. The solvent is evaporated at high temperatures and the particles are collected in the bottom of the spray dryer. The optimization of the hot air temperature will guarantee the drying of the microstructures. The optimization of operational parameters (feed flow rate, hot air flow rate, feed temperature, inner and outlet air temperature), accompanied by a convenient type of wall material, guarantees the necessary parameters for control of microstructure characteristics and allows the best encapsulation efficiency and yield [4,56]. It is an economical method of encapsulation in the food industry; however, high temperatures are used to eliminate the solvent, which can cause degradation of the thermo-sensitive compounds (Table 2) [57]. This process has been used to encapsulate vitamins, flavors, fat, oils, and carotenoids [4].

### 3.1. Spray Drying Parameters

The feed conditions such as feed flow rate, hot air rate, spray-dryer inlet and outlet temperatures, and type of atomizer have an influence on the physicochemical characteristics of the produced structures [28].

Feed temperature is an important factor in the spray-drying technique because it affects the viscosity of the feed and therefore the morphology of the prepared structures and their encapsulation efficiency. An increase in the feed temperature will allow the production of droplets with small sizes. However, excess high temperatures might cause degradation of the biocompounds.

It is crucial to optimize the inlet temperature and flow rate. For high air temperatures, it is important to guarantee a high solid concentration of solution in order to produce structures with high encapsulation efficiency while preserving the biocompounds.

The outlet temperature depends on the inlet temperature and feed rate. A high outlet temperature allows for a fast evaporation of the solvent and a quick formation of the wall structure, but it might degrade the activity of biocompounds. On the other hand, a low outlet temperature is necessary with an increased feed rate because there is a high amount of liquid in contact with the drying gas [56]. It is possible to reduce the number of wrinkled capsules by using low inlet temperatures during the spray-drying technique [58].

Several wall materials including carbohydrates such as maltodextrin, modified starch, cellulose derivates, gum and cyclodextrins, whey proteins, modified chitosan, and gelatin are used in this technique. Maltodextrin is the most popular polymer in the spray-drying technique to encapsulate compounds [59].

### 3.2. Encapsulation of Water-Soluble Vitamins by Spray Drying

Several studies regarding these vitamin encapsulation methods by spray drying have been reported in the literature. Some recent studies are reported in Table 3.

The bioavailability, stability, and non-toxicity are the main characteristics that depend on the type of biopolymers that allow the fabrication of suitable structures for food applications. For instance, Carlan et al., reported the efficient encapsulation of vitB1 (66–100%) using several types of wall biopolymers by spray drying and the results showed a product yield of 17–52% [35]. In addition, different vitB1 behaviors are presented: a fast release of the vitamin for systems based on biopolymers such as maltodextrin, modified starch, GA, and sodium alginate, and a slow release was obtained for chitosan and pectin microstructures.

In a comparative study, the encapsulation of folic acid using whey protein concentrate (WPC) and starch by the nanospray-drying technique was reported [57]. SEM photographs showed that folic acid presented higher encapsulation efficiency when entrapped by the produced WPC capsules than by the starch capsules. A comparative study of the folic acid release profiles was reported by Estevinho et al. [60]. Different encapsulating agents were used to entrap folic acid. The results showed regular spherical pectin microparticles and irregularly modified starch microparticles presented a high product yield (49.8% and 44.4%, respectively). The release of folic acid in deionized water (pH 5.6) was completed in less than two hours for all types of microparticles. It is reported that the starch system is a promising system for the encapsulation of folic acid [45]. In this study, starch and β-cyclodextrin were chosen as wall materials to protect vitB9 against acidic conditions in the stomach and to enhance B9 release in the second part of the small intestine. The encapsulation efficiency of folic acid in the β-cyclodextrin matrix is higher than within starch microparticles; nevertheless, the release profile of B9 showed similar results for both systems.

Spray drying has been used to preserve the thermo-oxidative stability of vitamin C encapsulated in spherical microparticles composed of sodium alginate and GA [66]. As a result, the stability of vitamin C up to 188 °C has been reported. Although both systems present 90% of vitamin C encapsulation efficiency, the 9.1 μm of GA microparticles revealed a high product yield of the nutrient.

In another study, the enhanced bioavailability of vitamin B12 when encapsulated into microcapsules constituted by maltodextrin, GA, and Hi-Cap was investigated [4]. Estevinho et al., reported encapsulated vitamin B12 within spherical cyanobacterial polymeric microparticles [61]. An increase in the product yield of the process was verified when the encapsulating agent was combined with GA. In addition, a decrease in the vitB12 time release was observed, suggesting more bioavailability for the digestive system. In a comparative study, the encapsulation of vitamin B12 using different shell materials by the spray-drying technique was stated [62]. The results showed spherical and smooth B12 microcapsules with a product yield between 27% and 50%. Additionally, at least 4 months of microcapsules’ shelf life was observed. In order to gain a better understanding of the absorption in the digestive system, a release profile of vitB12 loaded into modified chitosan microparticles was explored by Carlan et al. [64]. The smooth spherical vitB12-modified chitosan microparticles presented a product yield of around 57%. Further, a fast release of B12 was verified in simulated gastric conditions when compared with the vitB12 release at room temperature. The stability tests of vitamin B12 were observed for at least 6 months of storage. Another study suggested the production of wrinkled zein microcapsules with 8.32% *w*/*w* of vitB12 by the spray-drying technique [63]. In this study, the determined vitB12 encapsulation efficiency was 82%.

Furthermore, Chatterjee et al., investigated the potential anti-inflammatory activity of vitamin B1 and vitamin B6 entrapped in 4.5–4.8 μm ferulic acid-grafted chitosan microspheres [41]. A synergetic effect of vitB1 and vitB6-loaded microspheres was verified by the inhibition (52.3–56.9%) of carrageenan-induced paw edema in albino rats.

## 4. Electrohydrodynamic Techniques

Electrohydrodynamic techniques—electrospinning and electrospraying—are simple and versatile encapsulation techniques (Table 2) that have been used in several applications such as pharmaceutical, drug delivery, wound dressings, and tissue engineering. They allow the production of microstructures without significant toxic effects on the environment. In food and nutraceutical applications, these techniques were less studied but have been the focus of attention by the scientific community. They are known for their simplicity to fabricate fibers and particles with different physicochemical properties and different polymers with high scale-up potential [3]. These techniques present the capacity to fabricate structures and entrap hydrophobic and hydrophilic ingredients providing their stability (Table 2). This technology is focused on a high surface-to-volume ratio and efficient functionalized structures (Figure 4).

The main advantage of these techniques is that they do not require high temperature, pressure, or even the use of harsh chemicals, preserving the thermally labile biomolecules (Table 2, Figure 4) [44]. This technique presents scale-up potential and has the ability to fabricate structures with different polysaccharides such as chitosan, starch, and amaranth/pullulan; and protein-based structures such as zein and whey protein [3,44].

Figure 5 illustrates the general setup of electrospinning and electrospraying techniques.

The electrospinning technique involves a spinneret, a collector, and a high voltage source. Briefly, an applied voltage to the polymer solution induces a charged jet ejected from a Taylor cone (Figure 5A) [16]. The cone is achieved due to the absence of the balance of the electric forces and surface tension [69]. In the case of the electrospraying process, the polymer concentration is low enough to destabilize the jet and the two instabilities (Rayleigh and whipping) allow the formation of fine and highly charged droplets (Figure 5B). The former results from the surface tension of the solution which tends to minimize the surface area [70]. As a result, fibers and particles are collected, respectively. Factors such as the spinnability, production of fibers or particles, and diameter of the microstructures are affected by the solution parameters (polymer, viscosity, concentration and molecular weight of polymer, surface tension, solvents), processing variables (applied voltage, flow rate, diameter of needle, collector distance), and environmental parameters (temperature, humidity, airflow) [3]. These parameters can be adjusted in order to optimize the characteristics and morphology of the micro/nanostructures.

### 4.1. Electrospinning Factors

The polymer viscosity is closely related to the concentration, molecular weight, and polymer chain conformation [16]. In fact, viscosity has an influence on the solution spinnability. A high viscosity allows the production of fibers with large diameters and uniform morphology.

The increase in jet electrospinnability is directly related to the surface tension that occurs due to the applied voltage to the polymer solution [71]. Surfactants are frequently used for electrospinning fibers to increase the spinnability of the solution. At the same time, the use of surfactants allows for minimizing the size and homogeneity of the formed structures [56].

A solution with high electrical conductivity is also an essential parameter for the morphology and size of the fibers [56]. In fact, the charge density of the polymer has an influence on the elongation of the polymer jet. Commonly, high conductivity allows a high jet elongation causing a small diameter of the produced fibers.

The applied voltage is an important process parameter that provides surface charge on the polymer causing changes in the structure size. Usually, high applied voltage will promote small diameters due to the stretching of the polymer correlated to the electrostatic repulsive forces on the jet.

The formation of porous micro/nanostructures will be influenced by the used solvent. One other parameter that has an influence on the structures’ morphology and porosity is the flow rate [72]. Generally, the porosity and shape of the structures increase with high flow rates. The most used polymers are polysaccharides such as starch, pectin, and chitosan.

### 4.2. Type of Electrospinning

These techniques can occur in two different ways: by the coaxial electrohydrodynamic technique or by the direct incorporation of the compound within the polymeric solution.

The coaxial mode has advantages in microencapsulation processes because it can overcome the limitation associated with the effective entrapment of the ingredients with high encapsulation efficiency, high bioavailability, and monodisperse structures/formulations. The drawback correlated to the burst release of the encapsulated ingredients is minimized once the depositions of the ingredients do not occur on/near the structure’s surface [3]. Briefly, two concentric needles are used; one for the spinnable polymeric solution that is injected through the outer needle and the other for the encapsulated ingredients that are injected by the inner needle. The solution miscibility, evaporation, solvent diffusion, and surface tension between both liquid interfaces are important parameters related to the suitable formation of structures.

Structure fabrication by emulsion electrospinning can occur with water-in-oil (W/O) or oil-in-water (O/W) emulsion. The W/O emulsion is indicated for the entrapment of hydrophilic bioactive ingredients by hydrophobic polymers because there is protection of the ingredients from burst release and, at the same time, the ingredients’ bioactivity is assured.

### 4.3. Encapsulation of Water-Soluble Vitamins by Electrospinning and Electrospraying Techniques

A low number of systems have been developed to protect water-soluble vitamin stability using electrospinning and electrospraying techniques (Table 4).

For example, Ceylan et al., showed the possibility of using the electrospinning process to potentiate the VitB complex stability, entrapping it within chitosan nanostructures [32].

Aceituno-Medina et al., suggested amaranth:pullulan-based structures to entrap and photoprotect folic acid in order to be used in food fortification [73]. A high encapsulation efficiency (95.6%) was obtained and non-degradation occurred after 2 h of UV exposure. Additionally, entrapment enhanced the biocompound’s thermal stability.

In order to evaluate the folic acid thermal and irradiation resistance, the electrospinning/electrospraying process was used [44]. Folic acid was incorporated in zein microstructures, and an enhancement of the loaded folic acid thermal and UVA irradiation resistance was observed. In addition, folic acid was successfully entrapped by spray drying using a wall system based on starch as a promising food supplementation [43]. Fonseca et al., found that starch structures can potentially protect vitamins against thermal and UVA irradiation and during in vitro digestion (intestinal conditions) [43]. Pérez-Masiá et al., reported an improvement in the folic acid encapsulation efficiency as well as an increase in its stability when encapsulated into WPC capsules via the electrospinning technique [57].

Coelho et al., presented the production of zein microbeads with a size of around 3 μm as well as ribbon-shaped electrospun fibers with a vitB12 encapsulation efficiency above 90% [63]. These systems allow for the improvement of vitB12 bioaccessibility to be applied in food/nutraceutical products.

## 5. Physico-Chemical Characterization Techniques

### 5.1. Particle Size Distribution

Microstructures can be characterized by the laser diffraction technique. The size and percentage in the number and/or in volume of each is determined assuring precision of measurements.

### 5.2. Scannning Electron Microscopy (SEM)

This fast technique allows observing the size and shape of the structures.

### 5.3. Transmission Electron Microscopy (TEM)

The morphology of the microcapsules can be evaluated by TEM when their size is below typically around 200 nanometers.

### 5.4. Atomic Force Microscopy (AFM)

AFM is a crucial method for analyzing the mechanical properties of the structures via the obtained topographic surface images.

### 5.5. X-ray Diffraction (XRD)

This technique is used to analyze the crystalloid phase of the materials.

### 5.6. Fourier Transformed Infrared Spectroscopy-Attenuated Toral Reflectance (FTIR-ATR)

This technique is used to evaluate the microcapsule shell material. It is also possible to study the degradation of the microstructure’s external part.

### 5.7. Differential Scanning Calorimeter (DSC)

The stability and release profiles are important thermal-physical properties that can be measured by DSC. It measures endothermic and exothermic transitions as a function of the temperature. It provides the melting and solidifying enthalpy of a sample, important parameters for evaluating the stored energy capacity.

### 5.8. Thermogravimetrical Analysis (TGA)

This technique measures the change rate in the weight of materials as a function of temperature or time in a controlled atmosphere and evaluates their thermal stability.

### 5.9. Functional Characteristics

The functional characteristics of microstructures should be considered in the development of new matrices in food and nutraceutical areas.

The solubility analysis of microstructures is useful to determine their behavior in a solvent. This parameter has an influence on the encapsulation efficiency of the samples. The surface tension is a property of the material. The cohesive forces or the interfacial forces on the fluid membrane are responsible for this phenomenon. The hygroscopicity of a microstructure is important for observing its activity when exposed to an environment and, therefore, it has an influence on the stability of the samples. The encapsulation efficiency is determined based on the amount of encapsulated material and the theoretical amount of the material used. The highest or lowest efficiency depends on the core material concentration. The encapsulation efficiency decreases with the increase in the core material amount. The activity and stability of the bioactive ingredients are characteristics to evaluate. In fact, undesired factors such as heat, pH, light, etc., used in microencapsulation processes, must be controlled in order to guarantee the protection of the compounds. Controlled release is one way to evaluate the changes in the samples namely by using: a dissolution fluid, an osmosis pressure difference, enzymes to degrade the core, and a different pH to determine the influence on the solubility of the wall material.

## 6. Conclusions and Future Challenge

Spray-drying and electrohydrodynamic techniques are presented to the food industry as promising technologies for the loading and effective delivery of water-soluble vitamins. Different approaches are accessible for targeted vitamin delivery without changing their bioavailability. Several microstructures have been investigated due to their favorable physicochemical properties. The continuous search for structures for food and nutraceutical applications is a challenge. Based on the recent bibliography, polysaccharides are the most used for electrospinning and spray-drying processes.

Spray-drying is a simple and fast technique with high economic viability. It can protect sensitive vitamins and increase their stability against environmental factors. When compared with other technologies, spray drying presents a quick and easy scale-up methodology that makes it ideal in industrial applications.

Electrospinning is a technology that involves simple steps in comparison with other encapsulation techniques. It allows the fabrication of structures with different morphologies that exhibit many suitable characteristics in order to improve functional food products.

One of the limitations of a simple electrospinning/electrospraying setup is the low productivity due to the low solubility of the encapsulating agents associated with their poor viscoelastic behavior. Thereby, the multiaxial emulsion arrangement is one option for this technique. It can guarantee the production of large-scale structures by electrohydrodynamic techniques and promise environmental safety in the production of new formulations. Moreover, the release profile of a compound encapsulated via electrospinning tends to be quicker than via spray drying. The high control of environmental conditions (temperature and humidity) ensures reproducibility in the production process. Among these advantages, the non-use of temperature and the production of small diameter structures with high encapsulation efficiency make them important methods to encapsulate water-soluble vitamins for food and nutraceutical applications.

## Figures and Tables

**Figure 1 foods-11-01271-f001:**
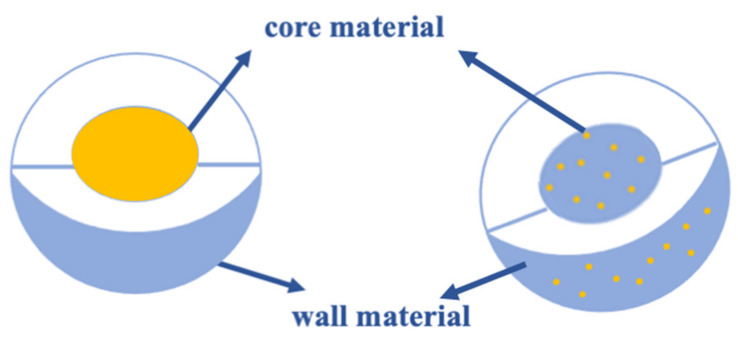
Morphology of microstructures.

**Figure 2 foods-11-01271-f002:**
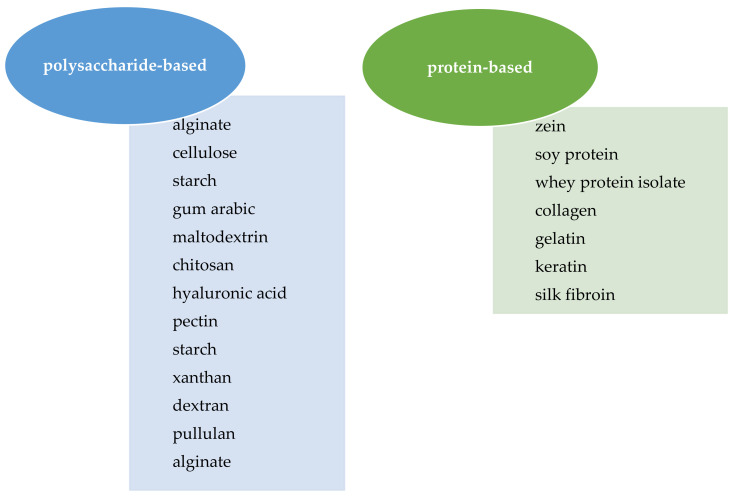
Examples of biopolymers used for microencapsulation processes.

**Figure 3 foods-11-01271-f003:**
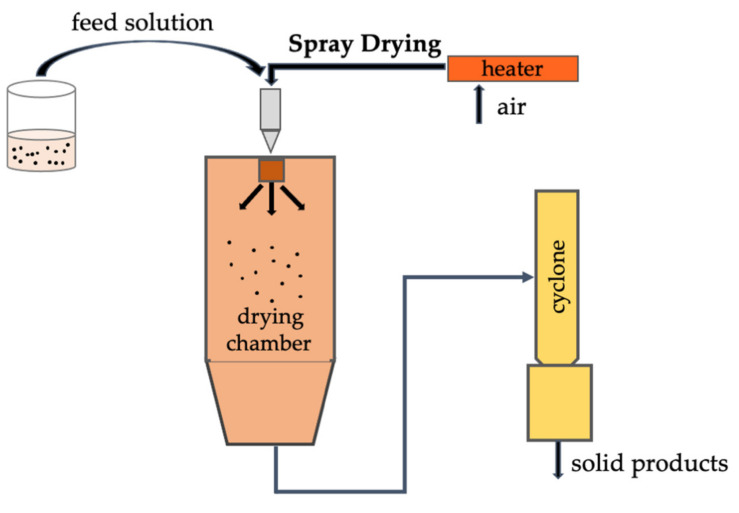
Schematic illustration of spray-drying setup.

**Figure 4 foods-11-01271-f004:**
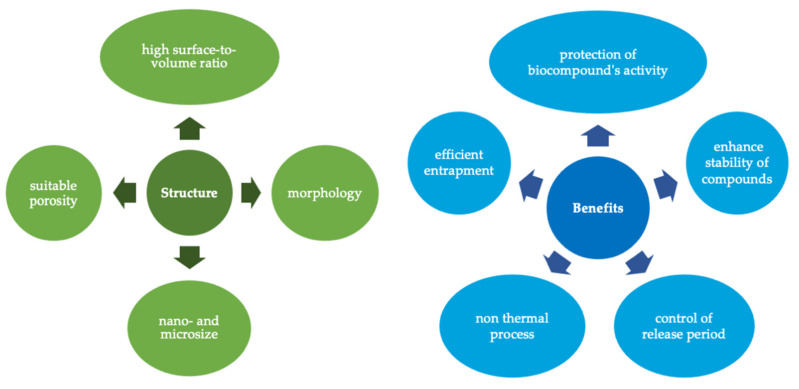
Advantages of electrospun/electrospray structures for food applications.

**Figure 5 foods-11-01271-f005:**
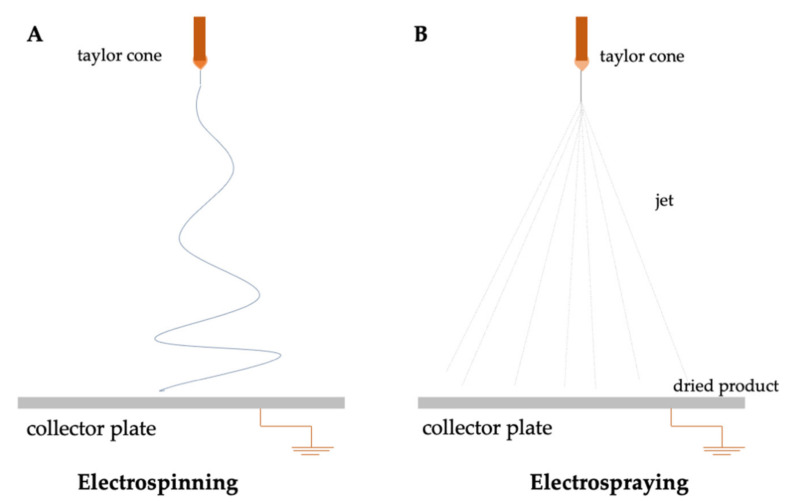
Schematic illustration of the basic setup for (**A**) electrospinning and (**B**) electrospraying.

**Table 1 foods-11-01271-t001:** Microencapsulation techniques [9,10,11].

Chemical Processes	Physical Processes
Coacervation	Spray drying
Interfacial polymerization Molecular Inclusion	Spray chilling Freeze drying
Co-crystallization	Melt extrusion Electrospinning Fluidized bed coating Solvent evaporation

**Table 2 foods-11-01271-t002:** Advantages and disadvantages of most common techniques used in food applications.

Encapsulation Technique	Advantages	Disadvantages
Complex coacervation [13,14]	Encapsulation of thermo-sensitive compounds Process at room temperature low cost	Use of toxic chemical solvents Residual solvent in the produced structures Medium/high cost-in-use
Electrohydrodynamic techniques [15,16]	Simple and versatile techniques efficient encapsulation Enhancement of biocompounds stability encapsulation of thermo-sensitive compounds Room temperature process	Difficult to scale up
Fluidized bed coating [17]	Low cost-in-use Homogeneity of a sample based on size.	Slow process High process cost Thermo-sensitive compounds degradation
Freeze-drying [18]	Long-term storage Large-scale production	Complex process Residual solvents in products High cost
Ionic gelation [19]	Mild conditions during the process Low cost	Low hydrophilic compounds encapsulation efficiency
Liposomes [14,20]	Encapsulation of aqueous/lipid soluble compounds Efficient controlled delivery	Laboratory scale Short half-life High cost
Melt extrusion [21,22]	Solvent free Continuous process Easy to scale up	Not recommended for thermolabile compounds High temperature and energy
Solvent evaporation [13]	Simple procedure Low cost-in-use	Low encapsulation efficiency
Spray chilling [23]	Low temperature in the process Low cost Easy to scale up	Changes in compounds activity due to fast cooling rates Low encapsulation efficiency low shelf life
Spray drying [13]	Simple procedure Short time of production Good encapsulation efficiency Good product stability Scale-up to commercial manufacture Continuous production Low cost-in-use Uniform spherical structures	High energy consumption Process at high temperature

**Table 3 foods-11-01271-t003:** Examples of the application of the spray-drying technique for water-soluble vitamins for food and nutraceutical applications.

Vitamin	Encapsulation Agent	Processing Parameters	Structures Average Size (µm)	Product Yield (%)	Encapsulation Efficiency (%)	Reference
Vitamin B1	Gum arabic Carrageenan Chitosan Maltodextrin Modified chitosan Modified starch Pectin Sodium alginate Xanthan	4 mL/min T_inlet_ = 120 °C T_outlet_ = 50–67 °C	0.11–1.32	17–52	66–100	[35]
	Chitosan and ferulic acid	10 mL/min T_inlet_ = 140 °C T_outlet_ = 77 °C	4.5–4.8	63.58–65.12	91 ± 2.31	[41]
Vitamin B6	Chitosan and ferulic acid	10 mL/min T_inlet_ = 140 °C T_outlet_ = 77 °C	4.5–4.8	63.58–65.12	83 ± 3.17	[41]
Folic acid	Gum arabic Modified chitosan Modified starch Pectin Sodium alginate	4 mL/min, T_inlet_ = 120 °C T_outlet_ = 58 °C	0.1–3.0	13.1–49.8	100% (except for modified starch)	[60]
	Cape gooseberry and maltodextrin	1.5 L/h T_inlet_ = 194.2 °C T_outlet_ = 87.7 °C	-	-	90.9 ± 1.8	[53]
	Starch β-cyclodextrin	~140 L/h T_inlet_ = 130 °C T_outlet_ = 80 °C	28.26–227.34 30.09–145.93	50.29 53.15	57.29 76.10	[45]
	Whey protein concentrate Starch	140 L/h T_inlet_ = 90 °C T_outlet_ = 45 °C	0.2–4.5	-	83.9 ± 7.8 52.5 ± 7.6	[57]
Vitamin B12	Gum acacia Modified starch Maltodextrin	T_inlet_ = 140 °C T_outlet_ = 60 °C	0.279–1.277	-	57.64–72.03	[4]
	Cyanobacterial extracellular polysaccharide, gum arabic	4 ml/min, T_inlet_ = 120 °C T_outlet_ = 65 °C	6–9	18.8	-	[61]
	Sodium alginate Carrageenam Maltodextrin Pectin Gum arabic Modified starch Xanthan	4 ml/min T_inlet_ = 120 °C T_outlet_ = 56–67 °C	0.93–2.74	27–50	-	[62]
	Zein	4 mL/min; Tin = 90 °C; Tout = 50 °C	2.23	83.1	82.3	[63]
	Modified chitosan	4 ml/min T_inlet_ = 120 °C T_outlet_ = 53–58 °C	3–8	56.0–58.0	-	[64]
Vitamin B12 Vitamin C	Chitosan, modified chitosan Sodium alginate	4 L/h T_inlet_ = 120 °C T_outlet_ ~ 65 °C	3	41.8–55.6 43.6–45.4	-	[48]
Vitamin C	Casein gel	0.54 L/h T_inlet_ = 180 °C T_outlet_ ~ 80 °C	5.8 ± 3.1	-	44.5 ± 1.2	[65]
	Sodium alginate Gum arabic	2–7 mL/min T_inlet_ = 140 °C T_outlet_ = 86 °C	9.1 6.0	74 51	90	[66]
	Cape gooseberry and maltodextrin	1.5 L/h T_inlet_ = 194.2 °C T_outlet_ = 87.7 °C	-	-	69.7 ± 0.7	[53]
	Sodium alginate	T_inlet_ = 110 °C T_outlet_ = 65 °C		30	93.48	[67]
	TPP-chitosan	3 L/h T_inlet_ = 175 °C T_outlet_ = 87.7 °C	8.0–9.0	61.1–62.8	45.05–58.30	[68]

-: Data not available.

**Table 4 foods-11-01271-t004:** Water-soluble vitamin encapsulation by electrospinning/electrospraying techniques for food and nutraceutical applications.

Vitamin	Encapsulation Agent	Processing Parameters	Encapsulation Efficiency (%)	Structures Average Size (µm)	Reference
Folic acid	Zein	1 mL/h, 16 cm,16 kV 0.6mL/h, 10 cm, 16 kV	92.9 98.6	0.70 (fibres) 0.27 (capsules)	[44]
	Starch	0.6 mL/h, 20 cm, 25 kV	73–95		[43]
	Whey protein concentrate Starch	0.15 mL/h, 9–11 cm, 10 kV	80.8 ± 12.9 44.0 ± 5.5	0.2–4.5	[57]
	Amaranth:pullulan	0.4 mL/h, 10 cm, 22 kV	95.6 ± 0.2	0.31–0.59	[73]
Vitamin B12	Zein	0.2 mL/h, 7 cm, 20 kV	100 91	0.31–0.5 (fibres) 1.25 to 4.38 (microspheres)	[63]

## Data Availability

Data that support the findings of this study are available on request to the authors.

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
