# Peer review of "Recent Advances in Water-Soluble Vitamins Delivery Systems Prepared by Mechanical Processes (Electrospinning and Spray-Drying Techniques) for Food and Nutraceuticals Applications—A Review"

_foods, 2022, doi:10.3390/foods11091271_

Round 1

Reviewer 1 Report

The paper titled: ,,Recent advances in water-soluble vitamins delivery systems prepared by mechanical processes (electrospinning and spray drying techniques) for food and nutraceuticals applications – a review” sent by Sílvia Castro Coelho, Berta N. Estevinho, F. Rocha describes different methods used for encapsulation of water-soluble vitamins.   

The article is very interesting, but I have a few suggestions.

Why the Authors' affiliation was written in Their mother tongue?

Table 2 is very good and has been carefully prepared, what is an important advantage of the article.

In my opinion in the chapter about vitamin B3 a few sentences about Pellagra should be added.

In the text I noticed a few mistakes for example: line 428 - Fonseca et al. or Coelho et al. what was the number of references?

The References have to be viewed thoroughly and corrected.

In my opinion the article, after minor revision, is suitable to be published in Foods.

The paper titled: ,,Recent advances in water-soluble vitamins delivery systems prepared by mechanical processes (electrospinning and spray drying techniques) for food and nutraceuticals applications – a review” sent by Sílvia Castro Coelho, Berta N. Estevinho, F. Rocha describes different methods used for encapsulation of water-soluble vitamins.   

The article is very interesting, but I have a few suggestions.

Why the Authors' affiliation was written in Their mother tongue?

Table 2 is very good and has been carefully prepared, what is an important advantage of the article.

In my opinion in the chapter about vitamin B3 a few sentences about Pellagra should be added.

In the text I noticed a few mistakes for example: line 428 - Fonseca et al. or Coelho et al. what was the number of references?

The References have to be viewed thoroughly and corrected.

Author Response

Suggestions accepted.

- Authors’ affiliation was modified in the manuscript.

- New sentence about pellagra condition was included in the “2.1.3. Vitamin B3” chapter, line 190/191: “. The vitB3 deficiency in the diet is the cause for pellagra nutritional disease [38].”

- The reference of line 465 was included in the text.

- References section was revised.

Reviewer 2 Report

The tables are different; please unify and use capital letters when necessary.
The figures must be substantially improved and also use capital letters when necessary.
In the abstract, the authors talk about the advantages and disadvantages; however, at no specific point in the review is there an important mention of both except for a table.
Mix abbreviations with full names; please correct or unify
Being a literature review, the number of references is scarce.
More observations are made in the attached document.

The studies used in this review need further discussion.

Author Response

Thank you for the suggestions.

- The tables were uniformized and capital letters included when necessary.

- Please see the new figures 1, 2, 3, 4 and 5 on the manuscript.

- The used abbreviations were verified and some sentences were corrected in the article.

- References section was revised.

- The manuscript was changed according to the observations reported by the reviewer.

Round 2

Reviewer 2 Report

The authors have taken into account and corrected the comments made to the manuscript.